# Computational Fluid Dynamics Simulations to Personalize Nasal Irrigations

**DOI:** 10.3390/jpm15070288

**Published:** 2025-07-03

**Authors:** Thomas Radulesco, Dario Ebode, Ralph Haddad, Jerome R. Lechien, Lionel Meister, Stephane Gargula, Justin Michel

**Affiliations:** 1IUSTI Laboratory, Centre National de la Recherche Scientifique (CNRS), Aix Marseille University, 13001 Marseille, France; thomas.radulesco@ap-hm.fr (T.R.); lionel.meister@univ-amu.fr (L.M.); stephane.gargula@ap-hm.fr (S.G.); justin.michel@ap-hm.fr (J.M.); 2ENT-HNS Department, La Conception University Hospital, 13005 Marseille, France; ralph.haddad@ap-hm.fr; 3Department of Anatomy and Experimental Oncology, Mons School of Medicine, UMONS Research Institute for Health Sciences and Technology, University of Mons, 7000 Mons, Belgium; jerome.lechien@umons.ac.be

**Keywords:** nose and paranasal sinuses, computational fluid dynamics, rhinology

## Abstract

**Background/Objectives**: Proper nasal irrigation techniques are essential for treating nasal and sinus conditions, influencing drug delivery efficiency and patient comfort. This study evaluates how different head positions—upright, right-tilted, and left-tilted—affect the distribution of saline solution in the nasal cavity and maxillary sinus using computational fluid dynamics (CFD). **Methods**: CFD simulations were conducted on a CT-based model of a healthy adult. A 4 mL saline solution was administered into the right nostril over three seconds. Fluid distribution and percentage of nasal mucosa coverage was analyzed in the inferior, middle, and superior thirds of the nasal cavity and the right maxillary sinus. **Results**: In the upright position, fluid primarily accumulated in the inferior (0.075 mL) and middle (0.015 mL) nasal regions, with minimal sinus penetration (0.002 mL). Right-tilting improved maxillary sinus coverage (0.028 mL) and increased irrigation of the inferior region (0.086 mL), while left-tilting enhanced central nasal coverage with only slight sinus penetration improvement. Irrigation patterns exhibited a rapid initial wetting phase followed by a slower, steady increase. **Conclusions**: Head position significantly influences the distribution achieved by nasal irrigation. These findings can guide clinical recommendations for specific conditions or postoperative care.

## 1. Introduction

Computational fluid dynamics (CFD) is the study of the flow of fluids, or their effect, through the numerical solution of the equations governing fluid behavior. Over the past decade, numerous studies focusing on CFD in the nasal airway have been published to better understand airflow and the processes of air conditioning and humidification [1]. These efforts have highlighted how geometrical and physiological factors—such as turbinate size, nasal valve angle, and sinus ostium patency—can substantially influence air movement and, consequently, patient comfort and health.

Today, CFD can also be applied to study the administration of nasal therapeutics, such as nasal irrigations or sprays [2,3]. These techniques aim to cleanse the nasal passages, deliver topical treatments, or address specific pathologies, including chronic rhinosinusitis or allergic rhinitis. However, effective nasal irrigation remains highly dependent on variables like fluid volume, flow rate, device design (e.g., nozzle diameter and angle), and head position. Despite multiple in vivo, in vitro, and in silico investigations, there is still no widely accepted consensus on the best way to perform nasal lavages. Disparate clinical recommendations and patient instructions underscore the need for more evidence-based guidelines. CFD is particularly well-suited for advancing personalized medicine in the context of nasal therapies because it enables patient-specific modeling. By generating three-dimensional reconstructions from a patient’s CT or MRI scans, clinicians and researchers can create anatomically accurate digital twins of that individual’s nasal passages. Any unique features, such as deviated septum, enlarged turbinates, or post-surgical cavities, are captured in the model, allowing for highly specific simulation scenarios.

Head position has emerged as a potentially significant parameter for improving the reach and effectiveness of nasal lavages. Tilting the head in various orientations harnesses gravity to direct the saline solution toward targeted anatomical regions [4,5]. Although intuitive, the precise impact of these positional changes on fluid coverage remains less understood. Some clinicians propose forward-leaning or lateral head tilts, while others recommend a more neutral or upright posture. As a result, patients receive mixed messages on how to position themselves during nasal irrigation.

In this preliminary in silico study, we investigated whether head position—upright, 45° right-tilted, or 45° left-tilted—modifies the area irrigated by a 4 mL nasal spray in both the nasal cavity and the maxillary sinus. By examining fluid coverage in key nasal subsites (inferior, middle, and superior thirds of the nasal fossa) and in the maxillary sinus, this study helps quantify how a simple intervention can potentially enhance therapeutic outcomes.

## 2. Materials and Methods

### 2.1. Ethical Considerations

The patient gave written consent before participating in the study. The study was conducted according to the guidelines of the Declaration of Helsinki, and approved by the AP-HM Institutional Review Board (N°2017-14-12-005) on 14 December 2017.

### 2.2. Guidelines

We followed EQUATOR guidelines.

### 2.3. Patient Selection

We included a healthy patient free of any nasal pathology or architectural malformation that might interfere with the procedure, whose CT scan data were available in our local database.

### 2.4. CFD Protocol

Three-dimensional reconstructions were obtained using ITK-Snap (3.6.0). The procedure was as follows: (1) importation of CT scan images (DICOM formats), (2) segmentation process using the half maximum height protocol (ImageJ software version 1.44) to determine the boundaries of anatomical structures, and (3) nasal surface extraction. CFD was performed using Star-CCM+^®^ software, version 2310 build 18.06.006 (Siemens^®^, Munich, Germany). Volume meshing was performed using a polyhedral mesher with parameters defined after a convergence mesh study: the boundary layer (total thickness = 0.16 mm) included 10 prismatic cells with a 1.1 prism layer stretching ratio [1]. We defined the following computational assumption: airflow is modeled under standard atmospheric pressure conditions (101,325 Pa). The relative pressure difference at the boundaries of the computational domain was considered to be zero. Air was considered to be an incompressible Newtonian fluid with density ρ = 1.225 kg/m^3^ and viscosity μ = 1.81 × 10^−5^ Pa·s. We considered the flow to be laminar as the device operated at a low speed (~1.5 m/s) and the estimated Reynolds number was ~1020.8. The sinonasal surface was a non-slip wall. Nasal lavages were performed using a nasal spray only on the right side with head upright, 45° right-tilted, and 45° left-tilted. Irrigations of 4.104 mL saline on the right nostril were simulated at a constant mass flow of 1.368 mL/s for 3 s. There was no inspiration or expiration simulated.

The simulations were performed using an implicit coupled second-order time scheme, with time steps of 10^−5^ s. This allowed for sufficient convergence for each solved equation (10^−3^). Liquid droplets were simulated using a Lagrangian solver, with the following physical properties: density 997 kg/m^3^ and dynamic viscosity 8.90 × 10^−4^ Pa·s. For each droplet, the Schiller–Naumann drag force model and the Sommerfeld shear lift coefficient were used [6]. The interaction of the droplets with the walls was simulated using a fluid film model with the Bai–Gosman wall impingement method [7].

In all head positions, we analyzed, for the right maxillary sinus, the upper, middle, and inferior third of the nasal fossa volumes reaching each area and the percentage of nasal mucosa irrigated. Head upright, we also analyzed the performance of the nasal spraying by analyzing the evolution of the spray’s volume-to-wetted-surface ratio and the wetted surface area.

## 3. Results

Simulations for the three head positions (upright, 45° right tilt, 45° left tilt) demonstrated substantial variation in how the saline solution dispersed within the nasal cavity and sinus (Figure 1). Table 1 details the volumes reaching each region, alongside the percentage of nasal mucosa irrigated. In the upright position, the fluid primarily targeted the inferior third, whereas tilting the head to the right markedly increased irrigation of the maxillary sinus. Conversely, the left tilt improved fluid reach in central areas of the nasal cavity, including the superior meatus and olfactory cleft.

Figure 2 and Figure 3 illustrate the spray volume-to-wetted-surface ratio and the wetted surface over time, respectively, for the upright head position. Early in the simulation, the solution exhibits limited effectiveness in wetting large areas—this is seen as a decreasing phase in the volume-to-wetted-surface ratio. However, around t = 0.07 s, the fluid becomes more efficient at coating fresh surfaces, shifting into an increasing phase. With time, as more areas become wet, the total wetted surface grows at a reduced rate, reflecting the diminishing opportunity to contact fully dry regions.

## 4. Discussion

### 4.1. Interest of CFD

Investigating the dispersion of topical irrigations into the sinus cavities has proven to be a complex task. Some research has explored irrigation distribution in both cadavers and patients, using various methods such as colored liquids, iodinated contrasts, and fluorescein labels [8,9]. However, these studies primarily focused on the final distribution and residual effects, neglecting the dynamics of irrigation flow. Predicting the precise extent of irrigation penetration remains a challenging aspect of these in vivo and ex vivo studies. Using CFD for nasal drug administration studies presents distinct advantages over in vivo experiments, allowing for a detailed exploration of factors like particle deposition, residence time, and regional variations. CFD simulations also enable researchers to optimize drug formulations and delivery devices by examining various nasal geometries and conditions. In essence, CFD circumvents many of the previous limitations by enabling detailed, step-by-step visualization of flow and droplet behavior. Researchers can systematically manipulate variables—such as head position, nozzle angle, or fluid properties—to predict their impacts on coverage. This approach can refine nasal drug administration, leading to more targeted deposition and potentially better therapeutic efficacy.

### 4.2. Impact of Head Position for Nasal Lavages

Initial spray application reaches a restricted anatomical area initially, but with high kinetics. A study of the timeline shows that the fluid film spreads over the wall in the direction of gravity as soon as its weight exceeds the surface tension force. The result is a high percentage of nasal mucosa reached in the lower third of the nasal cavity, especially around the inferior turbinate and in the lower part of the middle turbinate, especially if the head remains upright. This finding seems applicable to turbinal diseases such as rhinitis, or in post-operative conditions such as turbinoplasty or septoplasty.

Recirculation due to gas flow induced by particle movement and nozzle overpressure can spread the fluid film. Even when the patient is in an upright position, nasal lavage then reaches the upper third of the nasal cavity (upper meatus, spheno-ethmoidal recess at time > 1.5 s). Analysis of Figure 2 and Figure 3 revealed that initially, when the spray is introduced, its effectiveness in wetting the surfaces is limited (decreasing phase) until a specific time (~0.07 s) where the solution’s volume efficiently contributes to wetting the surfaces (increasing phase). Initially, surfaces are rapidly wetted, and as more areas become wet, the deposition process slows down, reflecting a diminishing rate of wetting for the remaining surface.

While these recirculating mechanisms could help to wash out sinus cavities after an operation, especially the maxillary sinus, the right- and left-tilted head simulations revealed a benefit in changing the head position to reach specific areas, as shown in other studies [3,10]. These data suggest that simple adjustments in head position may optimize nasal lavage for specific therapeutic objectives. For instance, washing the right nasal cavity with the head tilted to the right increases irrigation of the maxillary sinus. Patients with recalcitrant maxillary sinusitis may benefit from the right tilt (if the affected sinus is on the right side), potentially improving saline penetration into the sinus ostium. Washing the right nasal cavity with the head tilted to the left increases irrigation in the central area, particularly the olfactory cleft. This may be relevant to viral pathologies, in particular COVID-19, since this anatomical area has been described as highly endowed with ACE2 and TMPRSS2 receptors, responsible for the entry of the SARS-CoV-2 virus into the human body [11]. The upright position may suffice for conditions primarily affecting the inferior third, such as chronic rhinitis or mild turbinal hypertrophy. All data provided in this article support the efficiency of the nasal lavage at 3 s. Longer simulations would probably show improved results and should be performed in future studies.

### 4.3. Limits

We acknowledge several limitations to this work. Firstly, the simulations were carried out on a single case which, even if assumed to be healthy, cannot represent all the nasal conformations of healthy patients. These studies therefore need to be repeated on several anatomies, as well as on patients with nasal conditions, whether architectural (septal deviation) or inflammatory/infectious.

### 4.4. Future Development

This study represents a preliminary investigation conducted on a healthy subject. Future research will extend this model to include pathological and post-operative anatomies. By expanding these models, we aim to develop a personalized nasal irrigation protocol tailored to each patient’s specific condition and anatomical characteristics.

A promising future direction involves generating patient-specific CFD models that incorporate individualized anatomical features, such as a deviated septum, enlarged turbinates, or post-operative cavities. By customizing the boundary conditions and fluid properties to each patient’s scenario, it may be possible to predict the most effective irrigation angle, velocity, and even solution composition. The following are of particular interest:-Post-Operative Anatomies: Surgeries like septoplasty, turbinoplasty, or functional endoscopic sinus surgery (FESS) often modify the geometry of the nasal cavity, altering flow patterns. Personalized CFD can help tailor post-operative irrigation protocols to ensure optimal healing and reduce the risk of adhesions or bacterial colonization.-Pathological Conditions: Patients with nasal polyposis, chronic rhinosinusitis, or allergic rhinitis may present with inflamed, thickened mucosa that restricts airflow. CFD simulations could quantify how these inflammatory changes impact fluid spread, guiding more effective irrigation protocols.-Drug Formulation Studies: Beyond saline solutions, medicated sprays (e.g., steroids, antibiotics) vary in viscosity and surface tension. Patient-specific CFD might identify the ideal droplet size or concentration for enhanced mucosal residence time, particularly for areas prone to infection or inflammation

## 5. Conclusions

Head position emerges as a simple yet critical factor for maximizing the effectiveness of nasal irrigations. Our in silico data suggest that upright, right-tilted, and left-tilted head postures each uniquely influence fluid coverage in the nasal cavity and sinus regions. Although a single-subject study cannot fully capture the heterogeneity of nasal anatomies, these findings offer a basis for more refined, patient-centered approaches to nasal lavage. By refining CFD models and incorporating more complex anatomical variations, future work can pave the way toward truly individualized nasal irrigation strategies, enhancing both patient outcomes and overall treatment satisfaction.

Crucially, incorporating post-operative and pathological anatomies into these simulations will enable more precise irrigation protocols for patients with altered nasal structures—whether due to surgery, disease, or natural anatomical variation. Clinicians can leverage personalized CFD data to adapt lavage instructions to each patient’s unique needs, thereby improving therapeutic efficacy and potentially mitigating post-operative complications. As CFD technology and computational resources continue to advance, the prospect of near-real-time, patient-specific simulation becomes increasingly feasible, marking a significant step toward precision medicine in the realm of nasal and sinus care.

## Figures and Tables

**Figure 1 jpm-15-00288-f001:**
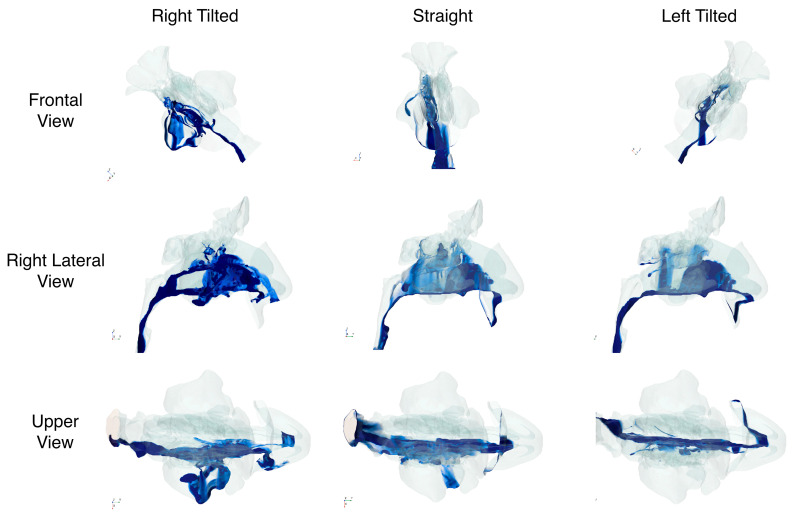
Frontal, right lateral, and upper views according to head position at time = 3 s.

**Figure 2 jpm-15-00288-f002:**
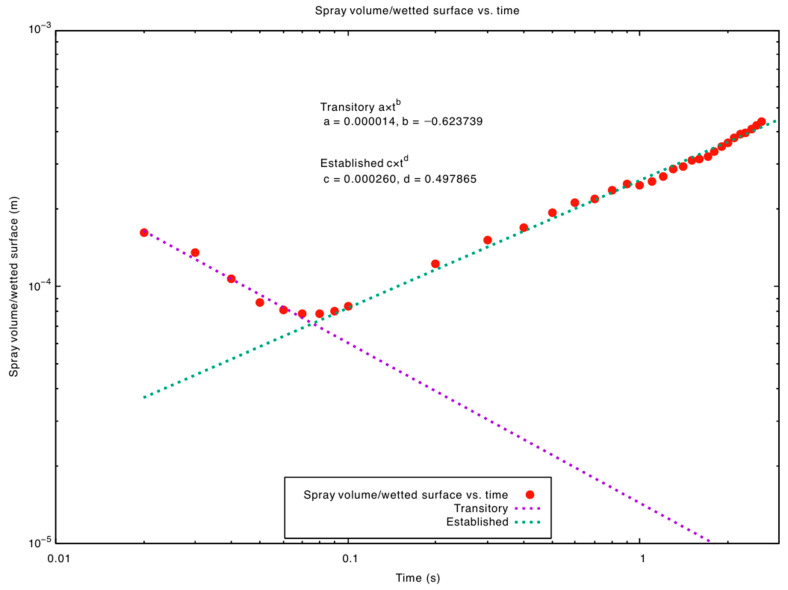
Evolution of the spray’s volume-to-wetted-surface ratio over time. It provides insight into the spray’s efficiency by showcasing its “performance” throughout different temporal phases. Initially, the spray is introduced but its effectiveness in wetting the surfaces is limited (decreasing phase) until a specific point (~0.07 s) where the solution volume efficiently contributes to wetting the surfaces (increasing phase).

**Figure 3 jpm-15-00288-f003:**
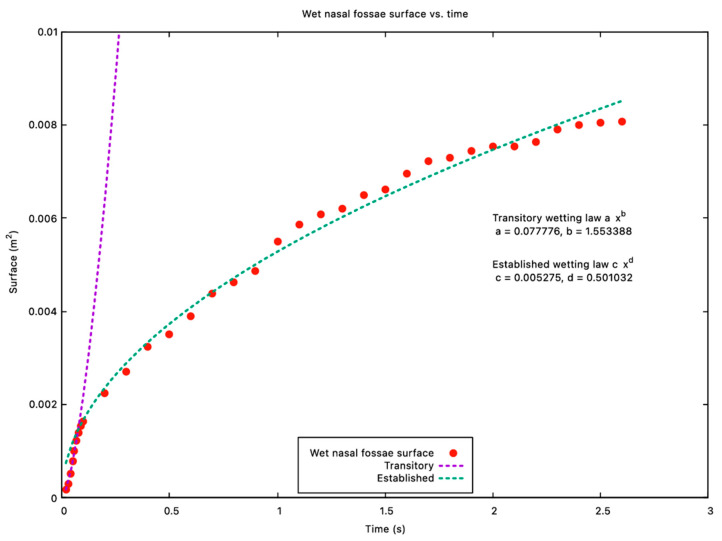
Wetted surface over time. This graph reveals two distinct phases in the spray process. The first phase demonstrates rapid growth, indicating the quick wetting of surfaces. In the second phase (after 0.07 s), the growth is slower, suggesting continued wetting but at a decreasing rate.

**Table 1 jpm-15-00288-t001:** Volumes and percentages of nasal mucosa irrigation function of head position.

Value	Head Position		
	Upright 0°	Right-Tilted 45°	Left-Tilted 45°
Volume reaching inferior third (mL)	0.75	0.86	0.52
Volume reaching middle third (mL)	0.15	0.22	0.09
Volume reaching superior third (mL)	0.05	0.02	0.01
Volume reaching right maxillary sinus (mL)	0.02	0.28	0.00
% of nasal mucosa irrigated (inferior third)	87	50	29
% of nasal mucosa irrigated (middle third)	63	41	30
% of nasal mucosa irrigated (upper third)	18	6	9
% of nasal mucosa irrigated (right maxillary sinus)	16	35	0.2
Total % of nasal mucosa irrigated	57	32	22

## Data Availability

Data is available on demand to the corresponding author.

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
