# Peer review of "Computational Fluid Dynamics Simulations to Personalize Nasal Irrigations"

_jpm, 2025, doi:10.3390/jpm15070288_

Round 1
Reviewer 1 Report
Comments and Suggestions for Authors
I couldn't understand the matierals and Methods, and whether the healty person was undertaken CT scan images, That is to say radiation. So if it's true than we can think that there is an ethical problem.
Author Response
Comment 1: I couldn't understand the matierals and Methods, and whether the healty person was undertaken CT scan images, That is to say radiation. So if it's true than we can think that there is an ethical problem.
Response: Thank you for your review. The CT scan images used to create our model were taken from a local anonymized database. While this particular scan appears normal, it was originally performed for clinical purposes unrelated to this study. No healthy volunteer was exposed to radiation for the purpose of this research.
The text has been modified as follows: “We included a healthy patient free of any nasal pathology or architectural malformation that might interfere with the procedure, whose CT scan data were available in our local database.” (line 78)
Reviewer 2 Report
Comments and Suggestions for Authors
CFD (computational fluid dynamics) is a widely applied study method in rhinology to determine the intranasal and sinus distribution of nasal topicals, the effect of endonasal surgery on the evolution of pathology. The endonasal distribution of irrigation solutions is a field of practical interest that has been approached so far on an intuitive basis. By using CFD this study manages to scientifically demonstrate the solution deposition modality in relation to head tilt, with particular practical implications. The limits and future perspectives of the research are also widely described. It is a fundamental study for any clinician interested in nasal diseases.
Comments on the Quality of English LanguageExcellent quality of the English language in this paper.
Author Response
Comment 1: CFD (computational fluid dynamics) is a widely applied study method in rhinology to determine the intranasal and sinus distribution of nasal topicals, the effect of endonasal surgery on the evolution of pathology. The endonasal distribution of irrigation solutions is a field of practical interest that has been approached so far on an intuitive basis. By using CFD this study manages to scientifically demonstrate the solution deposition modality in relation to head tilt, with particular practical implications. The limits and future perspectives of the research are also widely described. It is a fundamental study for any clinician interested in nasal diseases.
Response: Thank you for your kind review and the time spent on the manuscript.
Reviewer 3 Report
Comments and Suggestions for Authors
The manuscript requires major revisions to address several critical concerns regarding the CFD methodology and overall simulation setup. Please consider the following comments:
-
Temperature Setting (19°C / 292.15 K):
Why was 19°C selected as the operating temperature? Is there a clinical or experimental basis for this value? Please provide a reference or clear justification. -
Flow Regime and Turbulence Modelling:
The authors have modelled the flow as laminar. However, the Volume of Fluid (VOF) method typically deals with turbulent and complex multiphase flows, particularly in spray or high-velocity jet applications. Justification for using a laminar regime must be provided, especially considering the transient and potentially unstable nature of the flow. -
Multiphase Modelling Approach (DPM vs VOF):
It is unclear whether the Discrete Phase Model (DPM), Volume of Fluid (VOF), or another method was used for simulating the nasal irrigation. If DPM was used:-
What droplet/particle size was assumed?
-
Why was a particle-based method chosen, given that nasal irrigation generally involves continuous liquid flow rather than spray?
-
If a spray was simulated, what type of nasal irrigation device delivers 4 mL in 3 seconds? To the reviewer’s knowledge, typical irrigation devices (e.g., squeeze bottles) deliver significantly higher volumes over longer durations. If this was a spray simulation, what droplet breakup or tracking models were used? Also, clarify if the simulated scenario reflects a real-world device or is conceptual.
-
-
CFD Modelling Details:
The current description of the CFD setup is insufficient. The following must be included to ensure scientific rigour:-
A mesh independence test with results
-
A validation section, either against experimental data or from the literature
These are essential components of any CFD study.
-
-
Boundary Conditions – Inlet and Outlet Definition:
The manuscript does not clearly define the inlet and outlet boundary conditions. Is the outlet set at the nasopharynx or the contralateral nostril? From the figures, it appears the outlet is located at the nasopharynx. If so, this contradicts the typical bidirectional delivery method used in clinical nasal irrigation. The boundary conditions must be revisited and clarified. -
Overall Simulation Description:
The problem statement and CFD methodology lack sufficient detail. The modelling workflow, boundary settings, solver setup, and physical assumptions need to be described more clearly and in greater depth to ensure reproducibility and reliability of the results.
Author Response
- Temperature Setting (19°C / 292.15 K):
Why was 19°C selected as the operating temperature? Is there a clinical or experimental basis for this value? Please provide a reference or clear justification.
Response: Thank you for your comment. The selection of 19°C was made primarily to justify the air density and viscosity, as these properties are temperature-dependent. This value was chosen to provide a realistic reference for the airflow simulations. It is important to note that our simulation focuses solely on solving the Navier-Stokes equations without the thermal contribution.
The paragraph has been modified as follows: “We defined the following computational assumption: airflow is modeled under standard atmospheric pressure conditions (101,325 Pa)”
- Flow Regime and Turbulence Modelling:
The authors have modelled the flow as laminar. However, the Volume of Fluid (VOF) method typically deals with turbulent and complex multiphase flows, particularly in spray or high-velocity jet applications. Justification for using a laminar regime must be provided, especially considering the transient and potentially unstable nature of the flow.
Response: In natural, unforced breathing, the airflow in the nasal cavity is laminar. During the nasal lavage procedure, there is no inhalation or exhalation, and the device operates at a low velocity (around 1.5 m/s). The Reynolds number is approximately 1000, indicating that the flow is laminar and not turbulent. Therefore, a laminar flow model is considered to be the most appropriate choice for this scenario.
The text has been modified as follows: We considered the flow to be laminar as the device operated at a low speed (~1.5m/s) and the estimated Reynolds number was ~ 1020.8 (line 91)
- Multiphase Modelling Approach (DPM vs VOF):
It is unclear whether the Discrete Phase Model (DPM), Volume of Fluid (VOF), or another method was used for simulating the nasal irrigation. If DPM was used:- What droplet/particle size was assumed?
- Response: In this simulation, the Discrete Phase Model (DPM) was used, as stated in line 97. The droplets size distribution was measured by a company specializing in this field. However, due to industrial privacy concerns, we cannot disclose the droplet size in the manuscript.
- Why was a particle-based method chosen, given that nasal irrigation generally involves continuous liquid flow rather than spray?
- Response: In this particular model, we chose to study the effect of a continuous spray rather than a continuous liquid flow.
The text has been modified as follows:
- In this preliminary in silico study, we investigated whether head position—upright, 45° right-tilted, or 45° left-tilted—modifies the area irrigated by a 4 mL nasal spray in both the nasal cavity and the maxillary sinus. (l 64)
- Nasal lavages were performed using a nasal spray only on the right side with head up-right, 45° right-tilted and 45° left-tilted. (l.92)
- If a spray was simulated, what type of nasal irrigation device delivers 4 mL in 3 seconds? To the reviewer’s knowledge, typical irrigation devices (e.g., squeeze bottles) deliver significantly higher volumes over longer durations. If this was a spray simulation, what droplet breakup or tracking models were used? Also, clarify if the simulated scenario reflects a real-world device or is conceptual.
Response: The data used for the simulation was obtained from measurements provided by the industry. The simulation models a nasal spray with saline solution, rather than a flow-type lavage, using a Probability Distribution Function (PDF) to model the droplet behavior. This spray is a mist made up of very fine droplets spaced far apart from each other (no accretion can be observed until droplets touch the nasal fossae).
- CFD Modelling Details:
The current description of the CFD setup is insufficient. The following must be included to ensure scientific rigour:- A mesh independence test with results
- A validation section, either against experimental data or from the literature
These are essential components of any CFD study.
Response: The CFD model has been previously validated in multiples study, and in particular the reference [1], ensuring its reliability and accuracy.
The text has been modified as follows: The reference [1] has been added to the text to give more context to this section. (line 89)
- Boundary Conditions – Inlet and Outlet Definition:
The manuscript does not clearly define the inlet and outlet boundary conditions. Is the outlet set at the nasopharynx or the contralateral nostril? From the figures, it appears the outlet is located at the nasopharynx. If so, this contradicts the typical bidirectional delivery method used in clinical nasal irrigation. The boundary conditions must be revisited and clarified.
Response: No inspiration or expiration occurs during the simulation. Since there is no airflow or pressure difference at the boundaries of the computational domain, the position of the inlet/outlet does not impact the model. This is in line with the spray process conditions, where no breathing cycle is involved during the irrigation.
The text has been modified as follows: “Relative pressure difference at the boundaries of the computational domain was considered to be zero.” (l.89)
- Overall Simulation Description:
The problem statement and CFD methodology lack sufficient detail. The modelling workflow, boundary settings, solver setup, and physical assumptions need to be described more clearly and in greater depth to ensure reproducibility and reliability of the results.
Response: Thank you for your review. We hope we were able to add enough information to the method section to provide greater understanding.
Round 2
Reviewer 3 Report
Comments and Suggestions for Authors
Accepted